# Pea Protein-Derived Peptides Inhibit Hepatic Glucose Production via the Gluconeogenic Signaling in the AML-12 Cells

**DOI:** 10.3390/ijerph191610254

**Published:** 2022-08-18

**Authors:** Wang Liao, Xinyi Cao, Hui Xia, Shaokang Wang, Guiju Sun

**Affiliations:** Key Laboratory of Environmental Medicine and Engineering of Ministry of Education, and Department of Nutrition and Food Hygiene, School of Public Health, Southeast University, Nanjing 210009, China

**Keywords:** pea protein hydrolysate, hepatic glucose production, gluconeogenic signaling, insulin signaling

## Abstract

Pea protein is considered to be a high quality dietary protein source, but also it is an ideal raw material for the production of bioactive peptides. Although the hypoglycemic effect of pea protein hydrolysate (PPH) has been previously reported, the underlying mechanisms, in particular its effect on the hepatic gluconeogenesis, remain to be elucidated. In the present study, we found that PPH suppressed glucose production in mouse liver cell-line AML-12 cells. Although both of the gluconeogenic and insulin signaling pathways in the AML-12 cells could be regulated by PPH, the suppression of glucose production was dependent on the inhibition of the cAMP response element-binding protein (CREB)-mediated signaling in the gluconeogenic pathway, but not the activation of insulin signaling. Findings from the present study have unveiled a novel role of PPH underlying its anti-diabetic activity, which could be helpful to accelerate the development of functional foods and nutraceuticals using PPH as a starting material.

## 1. Introduction

Type 2 diabetes mellitus (T2D) is considered to be a global pandemic, and its associated complications significantly contribute to the increase of disability and mortality, which pose a great burden on the global healthcare system. It has been estimated that there are 400 million diabetic patients globally, and T2DM accounts for about 95% of these cases [1]. The increasing prevalence of T2DM and its subsequent socioeconomic burden encourages new strategies to manage T2DM.

The pathophysiology of T2DM has been discussed for decades, and a universal consensus is that hyperglycemia resulting from defective insulin action and/or hyperglucagonemia causes the onset of T2DM [2]. The liver is a major organ that responds to insulin and glucagon, which executes the functions of the endocrine hormones in control of glucose production [3]. In the fasting state, glucagon degrades macromolecules and stimulates hepatic gluconeogenesis. However, insulin plays a role in the liver to suppress hepatic glucose production (HGP). Glucagon and insulin reciprocally control metabolic homeostasis, which requires a tight control of gene transcription to exert the opposing effects of these two hormones [4]. Hereby, the modulation of HGP has been considered to be an essential strategy for the management of T2DM [5].

Many studies have shown that the prolonged use of hypoglycemic drugs is associated with certain side effects, such as hypoglycemic reactions and adverse gastrointestinal reactions. In addition, long-term drug use is often costly, resulting in an economic burden [6]. Compared with the synthetic hypoglycemic drugs, natural compounds have advantages with fewer side effects and have a relatively low production cost. Therefore, dietary supplementation with natural compounds for the intervention of T2DM has gained substantial attraction. The compounds that have been reported to have a hypoglycemic effect include phytochemicals, dietary fiber, polyunsaturated fatty acids, and food protein-derived bioactive peptides [7]. Of note, clinical trials of bioactive peptides for the management of hyperglycemia have been reported. Randomized controlled trials of peptides derived from the proteins of milk, eggs, and seafoods have been designed and the hypoglycemic effects in T2DM subjects have been documented [8].

Pea (*Pisum sativum* L.) is one of the most important pulse crops that covers more than one third (34.2%) of the total area of land used to grow dry pulses [9]. Pea protein accounts for about 20% of the dry basis of pea flour and is considered to be a high-quality product [10]. The pea proteins are rich in essential amino acids which are deficient in cereals proteins. In addition, the digestibility-corrected amino acid score of pea protein isolates was found to be higher than that of soy protein isolates [11,12]. In such context, pea protein has been used as a raw material for the production of bioactive peptides. Various physiological functions of pea protein-derived bioactive peptides have been reported, which indicates great potential for pea protein-derived bioactive peptides in the management of chronic diseases [13,14]. Although the anti-diabetic activity of pea protein-derived bioactive peptides have been documented [15], the underlying mechanisms remain to be elucidated.

Given the essential role of abnormal HGP in the development of T2DM, this study aimed to investigate whether pea protein-derived bioactive peptides could mitigate HGP by regulating glucagon and/or insulin signaling pathways.

## 2. Materials and Methods

### 2.1. Peptide Profiling of PPH

The PPH sample was subjected to the use of a nano-UPLC (EASY-nLC1200) coupled with the use of a Q Exactive HFX Orbitrap instrument (Thermo Fisher Scientific, Waltham, MA, USA) with a reversed-phase column (100 μm ID × 15 cm, Reprosil-Pur 120 C18-AQ, 1.9 μm) for separation based on a previous study, with modifications [16]. Acetonitrile (2% ACN) with 0.1% formic acid (FA) in H_2_O and 80% ACN with 0.1% FA in H_2_O were used as mobile phase A and phase B, respectively. Separation of the sample was conducted at 300 nL/min flow rate. The gradient B was set up at: 2–5% for 2 min, 5–22% for 34 min, 22–45% for 20 min, 45–95% for 2 min, and 95% for 2 min.

The Orbitrap analyzer was used to conduct data dependent acquisition (DDA) in the positive mode. The resolution was set up to 120,000 (@200 *m*/*z*) and 15,000 for the primary (MS1) and secondary (MS2) mass spectrometry tests, respectively. The automatic gain control (AGC) target for MS1 and MS2 was set to 1E6 and 1E5, respectively. The top 20 most intense ions were fragmented with 30% of the normalized collision energy. Single charged peaks and peaks with charges of more than 6 were not qualified for the DDA procedure.

The MS files were processed using the Proteome Discoverer (PD) software (Version 2.4.0.305, Thermo Fisher Scientific) and the built-in Sequest HT search engine. The sequences were searched for in the UniProt FASTA databases (uniprot-Pisum sativum_3888.fasta). The false discovery rate was set to 0.01. A specific peptide was identified using an initial precursor mass deviation of up to 10 ppm and a fragment mass deviation of 0.02Da.

### 2.2. Cell Culture

The mouse liver cell-line AML-12 samples were purchased from Guangzhou Saiku Biotechnology Co., Ltd. (Guangzhou, China) The cells were maintained in a DMEM/F12 medium supplemented with 10% fetal bovine serum, 0.005 mg/mL insulin, 0.005 mg/mL transferrin, 5 ng/mL selenium, 40 ng/mL dexamethasone, and penicillin-streptomycin and were kept at 37 °C in a 5% CO_2_ atmosphere until the confluence reached 80% for the further experiments.

### 2.3. Cell Viability

The cell viability was evaluated using a CCK-8 assay. The AML-12 cells were seeded in a 96-well plate and treated with different concentrations of PPH for 24 h. Afterward, the CCK-8 assay was conducted according to the assay kit protocol (APExBIO, Houston, TX, USA). The absorbance of the treated group at 450 nm was normalized to the corresponding control group, which represented the cell viability.

### 2.4. HGP Assay in AML-12 Cells

The HGP assay was conducted according to a previous study, with modifications [17]. AML-12 cells were seeded in a 6-well plate. The confluent cells were treated with PPH in serum-free DMEM for 1 h. Then, the cells were washed with PBS twice and co-treated with PPH and glucagon in glucose and phenol red free DMEM containing 2 mM sodium pyruvate and 20 mM sodium lactate for 3 h. The glucose concentration of the cell culture media was measured using the Amplex Red Glucose/Glucose Oxidase Assay Kit (Invitrogen, Waltham, MA, USA). The absorbance of the reaction mixture at 585 nm was measured using a spectrophotometer (BioTek Instruments, Inc., Winooski, VT, USA).

### 2.5. Western Blotting

The cells were seeded in a 6-well plate until confluence. The cells were washed with PBS and pre-treated with PPH for 1 h in the serum-free DMEM, followed by the co-treatment with glucagon for 6 h. After the treatment, the cells were lysed with the RIPA buffer (GenStar, Beijing, China) with protease and phosphatase inhibitors. These cell lysates were then run in 9% sodium dodecyl sulfate polyacrylamide gel electrophoresis (SDS-PAGE), transferred to a polyvinylidene fluoride membrane, and immunoblotted with the specific antibody. The anti-rabbit or anti-mouse antibody, which was used as the secondary antibody, was purchased from Santa Cruz. The protein band was exposed by a Tanon-5200 ECL detection reagent and the density of the bans were analyzed using the Image J software (https://imagej.nih.gov/ij/).

### 2.6. qRT-PCT

The cells were cultured and treated in the same way as is described above in the western blotting section. The Trizol reagent (Invitrogen) was used to extract total RNA from the cells, followed by the cDNA synthesis using the reverse transcription kit (Cat: RR036A, Takara Bio, Kusatsu, Shiga, Japan). A Real-Time PCR System (CFX Connect, Bio-Rad, Hercules, CA, USA) was used for the amplification. The relative mRNA levels were calculated by the equation 2^−ΔΔCt^ and presented as the fold change normalized to the internal control GAPDH. The sequence of the primers are as follows: G6PC (forward-5′CATTGTGGCTTCCTTGGTCC-3′; reverse-5′GGCAGTATGGGATAAGACTG-3′), PCK1 (forward-5′ CCATCGGCTACATCCCTAAG-3′; reverse-5′ GACCTGGTCCTCCAGATACC-3′), GADPH (forward-5′ CACCCCATTTGATGTTAGTG-3′; reverse-5′ CCATTTGCAGTGGCAAAG-3′).

### 2.7. Statistical Analysis

Data were presented as mean ± SEM (standard error of mean) of at least 3 independent experiments. A one-way analysis of variance (ANOVA) with the Tukey’s post-hoc test was used for the determination of statistical significance using PRISM 8 statistical software (GraphPad Software, San Diego, CA, USA). *p* < 0.05 was considered to be statistically significant.

## 3. Results

### 3.1. Characteristics of the Peptide Profile

The peptide profile is shown in Figure 1A. In total, 759 peptides were identified from the PPH (Appendix A), among which, RPVKEL had with the smallest molecular weight (741.5 kDa). RGDTIKIPA is the peptide with the highest abundance. Molecular weight distribution showed that the molecular weight of most of the peptides is within the range of 1000 to 1499 (Figure 1B). There were 188 oligopeptides with less than 10 amino residues (Figure 1C).

### 3.2. PPH Did Not Show Cytotoxicity in the AML-12 Cells

Cell viability was evaluated via the CCK-8 assay. PPH with the concentrations of 0.1, 0.2, 0.5, or 1.0 mg/mL were treated to the cells for 24 h. The cell viability was slightly increased after the treatment but with no statistical significance (Figure 2). Such a result indicates that PPH is not cytotoxic to the AML-12 cells. Based on a previous study [18], 0.2 and 0.5 mg/mL of PPH were used for further experiments.

### 3.3. PPH Suppressed Glucagon-Induced HGP in the AML-12 Cells

To assess the effect of PPH on the glucose production of AML-12 cells, the HGP assay was conducted. As expected, glucagon stimulation significantly improved the HGP in the AML-12 cells. The treatment with 0.2 mg/mL or 0.5 mg/mL PPH normalized the HGP in AML-12 cells, but there was no difference between the different concentrations. Thus, 0.2 mg/mL of PPH was used for further experiments. In addition, neither 0.2 mg/mL nor 0.5mg/mL PPH treatment affected the HGP in AML-12 cells significantly (Figure 3).

### 3.4. PPH Regulated Glucagon Signaling in the AML-12 Cells

Since PPH could inhibit the glucagon-induced HGP in AML-12 cells, we then explored its underlying mechanisms. CREB plays a pivotal role in glucagon signaling. The phosphorylation of CREB at S133 activates the transcription of gluconeogenic genes, such as G6PC and PCK, which is followed by glucose production. As shown by Figure 4A, glucagon stimulation significantly enhanced the phosphorylation of CREB at S133 by 5.6-fold, which decreases after the PPH treatment to 2.6-fold (Figure 4A,B). Consistently, the glucagon-induced transcriptions of G6PC and PCK are suppressed by the PPH treatment, which is evident by the reduced mRNA levels of these gluconeogenic genes (Figure 4C,D).

### 3.5. PPH-Activated Insulin Signaling in the AML-12 Cells

In addition to the glucagon signaling, we also investigated whether insulin signaling could be regulated by the PPH treatment. The insulin-stimulated phosphorylation of Akt at S473 (1.9-fold versus 3.0-fold) and Foxo1 at S253 (1.8-fold versus 2.8-fold) are further enhanced by the PPH treatment (Figure 5A,C,D). The expression of IRS1, which is found in the upstream of the insulin signaling pathway, is upregulated by the PPH treatment as well (Figure 5A,B). Interestingly, the phosphorylation of Foxo1 at S253 was improved by the PPH treatment alone, and the effect was comparable to that of insulin (Figure 5A,C).

### 3.6. PPH Suppressed the Glucose Production in AML-12 Cells via the Glucagon Signaling but Not the Insulin Signaling

Since both glucagon and insulin signaling were involved in the activity caused by the PPH in the AML-12 cells, we further employed the CREB inhibitor (KG-501) and Akt inhibitor to delineate the specific roles of these two pathways in mediating the effects of PPH. As shown by Figure 6A, the effect of PPH was negated by the addition of KG-501, which indicates that the suppression of HGP by PPH in AML-12 cells is dependent on glucagon signaling. On the contrary, despite the addition of the Akt inhibitor, the PPH treatment showed a significant effect in decreasing the HGP in AML-12 cells (Figure 6B), which suggests that the HGP inhibitory effect of PPH in AML-12 cells was not attributed to the activation of insulin signaling.

## 4. Discussion

Over the past few years, plant-derived proteins have been considered as preferred alternatives to animal proteins, owing to the growing concerns of their impact on health and environments [19]. Pea protein has a high nutritional value without having any genetic modification, and is also labelled as a non-allergic food substance [20]. Herein, it is an ideal raw protein for food development and production. PPH is one of the major by-products of pea protein. PPH causes various physiological functions including the hypoglycemic activity that has been reported [21,22,23]. To further elucidate the underlying mechanisms of PPH in reducing blood glucose, we investigated the effects of PPH on HGP in mouse liver cell-line AML-12 cells. Major findings include: (1) PPH inhibits glucagon-stimulated HGP in the AML-12 cells; (2) PPH regulates both gluconeogenic and insulin signaling pathways in the AML-12 cells; (3) PPH inhibits HGP dependent on the gluconeogenic signaling but not the insulin signaling in the AML-12 cells.

The metabolic modulatory effects of PPH have been documented. In the 3T3-L1 murine pre-adipocytes, PPH increased the mRNA levels of adiponectin and insulin-responsive glucose transporter 4 and promoted glucose uptake. Moreover, the expression of peroxisome proliferator-activated receptor γ is also be upregulated by the PPH treatment during the adipocyte differentiation. These findings indicate the health benefits of PPH for the control of obesity-associated metabolic disorders [24]. Of note, the effects of PPH for the control of blood glucose have been reported in both mice and humans. The administration of PPH improved glucose tolerance and glycogen synthesis in T2DM mice [25]. In healthy adults, the consumption of pea protein reduced postprandial glycemia and stimulated insulin release [26]. Since the pea protein could be digested by the gastrointestinal enzyme into protein hydrolysate, such a finding further showed evidence of the benefits of PPH for the control of blood glucose. In the present study, we reported that PPH could suppress HGP in the AML-12 cells, which unveiled a novel mechanism of PPH in glycemic control. 

In this study, we found that the inhibitory effect of PPH on HGP in the AML-12 cells was dependent on the gluconeogenic signaling. The regulatory effects of natural compounds on gluconeogenic pathways in hepatocytes have been reported previously. The red pepper seed extract treatment decreased the mRNA levels of G6PC and PCK1 and suppressed glucose production in AML-12 cells [17]. However, whether the inhibition of HGP was caused by the regulation of the gluconeogenic pathway was unclear. It was also reported that glycosaminoglycan from *Apostichopus japonicus* could alleviate the HGP by targeting the CREB-mediated signaling in the insulin resistant hepatocytes [27]. Additionally, the treatment of epigallocatechin gallate, which is a phytochemical, could suppress HGP in mice hepatocytes by antagonizing the protein kinase A/Foxo axis [28]. The findings from the present study showed the feasibility that the hepatic gluconeogenic signaling could be targeted by food protein-derived bioactive peptides, which creates a new opportunity for the development of gluconeogenesis targeting substance in control of blood glucose. Of note, the specific target of PPH and other bioactive peptides, if any, in the gluconeogenic signaling needs to be characterized by a gene knockout mice model in the future studies. 

The treatment of PPH was also found to regulate the insulin signaling in our study, which is consistent with a previous study reporting that the intragastric administration of PPH upregulated the gene expressions of IRS1 and IRS2 in the liver of diabetic mice, and reduced insulin resistance [16]. In addition, the ingestion of PPH enhanced the postprandial insulin secretion in the healthy subjects [29]. However, we found that the regulation of the insulin signaling did not directly contribute to the HGP-suppressing effect of PPH in the AML-12 cells. Thus, the beneficial role of PPH via activating the insulin signaling in the hepatocytes need to be further determined.

Although we profiled the PPH sample used in this study and 759 peptides were identified, we did not further purify and fractionate PPH, since the objective of this study was to evaluate the effect of PPH on HGP and its associated mechanisms and was aimed at providing a rationale for the animal and clinical studies in the future. Of note, LRW is a tripeptide originally identified as an inhibitor of angiotensin-converting enzyme 1 from pea protein [30], but recent studies also reported its activity in modulating the functions of vascular cells [31] as well as showing osteoblastic activity in bone cells [32]. Interestingly, LRW was found to be incorporated in three peptides identified from this study including DLPVLRWLK, DLPVLRWLKLSAE, and DLPVLRWLKLSAEHGSLH. Thus, we speculated that LRW might be the active sequence in these peptides, which could be released by the cells and could exert the effect of suppressing HGP in the AML-12 cells. On the other hand, the purification and fractionation of PPH needs to be considered in future studies to identify the key peptides of PPH in inhibiting HGP, which may be followed by a structure and function study to delineate the relationship between the structure and activity of the peptides.

## 5. Conclusions

In conclusion, the treatment of PPH could inhibit glucose production in the AML-12 cells via the gluconeogenic signaling pathway, suggesting the potential role of PPH in suppressing HGP. Findings from the present study delineate a novel role of PPH in exerting its anti-diabetic activity, which could be helpful to accelerate the development of functional foods and nutraceuticals using PPH as a starting material.

## Figures and Tables

**Figure 1 ijerph-19-10254-f001:**
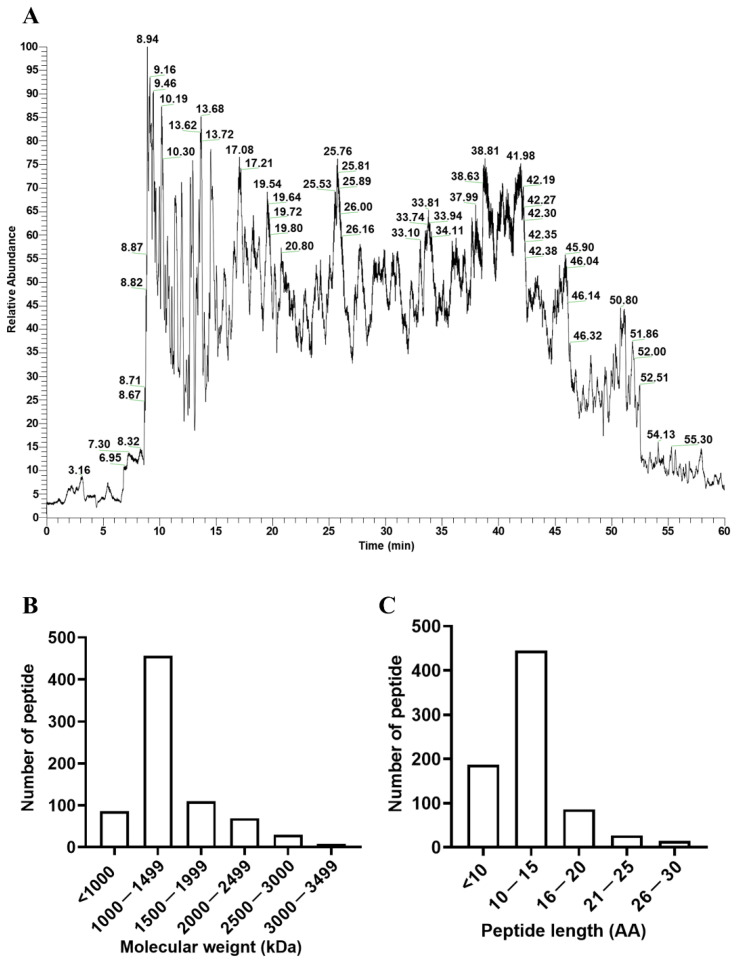
**Characteristics of the PPH peptide profile.** (**A**) The peptides in PPH were characterized using LC-MS/MS. (**B**) Molecular weight distribution of the identified peptides. (**C**) Peptide length distribution of the identified peptides.

**Figure 2 ijerph-19-10254-f002:**
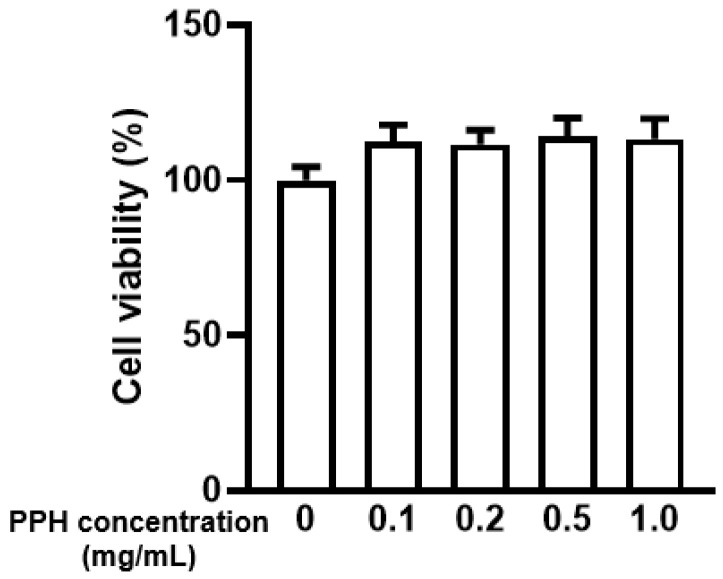
**PPH did not show cytotoxicity in the AML-12 cells.** AML-12 cells were treated with different concentrations of PPH for 24 h. The cell viability was evaluated using the CCK-8 assay. Data are expressed as mean ± SEM of 5 independent experiments.

**Figure 3 ijerph-19-10254-f003:**
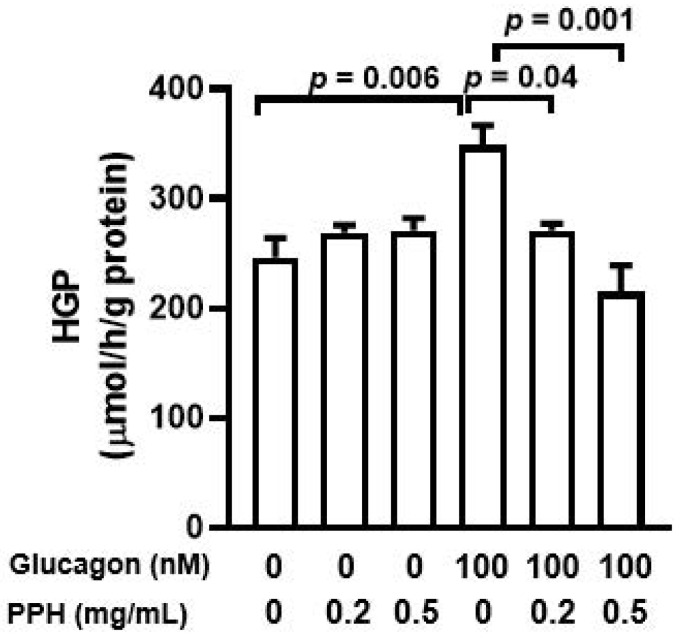
**PPH suppressed glucose production in the AML-12 cells.** AML-12 cells were cultured in glucose and phenol red-free DMEM containing sodium pyruvate and sodium lactate and pre-treated PPH, followed by the co-treatment with glucagon (100 nM) for 6 h. The concentration of glucose produced by the cells was normalized by the total protein content of the cells. Data are expressed as mean ± SEM of 5 independent experiments.

**Figure 4 ijerph-19-10254-f004:**
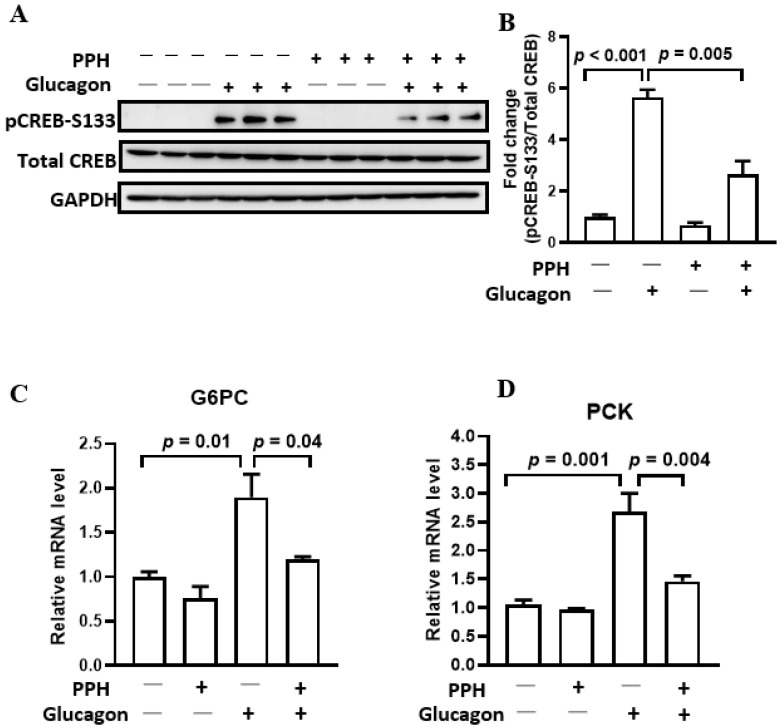
**PPH regulated glucagon signaling in the AML-12 cells.** (**A**) AML-12 cells were pre-treated with PPH for 1h, followed by the co-treatment with 100 nM of glucagon for 1h. The total protein of the cells was extracted and immune-blotted with the antibodies against pCREB-S133, total CREB, or GAPDH. (**B**) The protein band of pCREB-S133 was quantified and normalized to the total CREB. (**C**,**D**) AML-12 cells were pre-treated with PPH for 1h, followed by the co-treatment with 100 nM of glucagon for 6 h. Total RNA of the cells were extracted. The mRNA level of G6PC and PCK were assessed using RT-PCR, and GAPDH was used as the internal control. Data are expressed as mean ± SEM of at least 3 independent experiments.

**Figure 5 ijerph-19-10254-f005:**
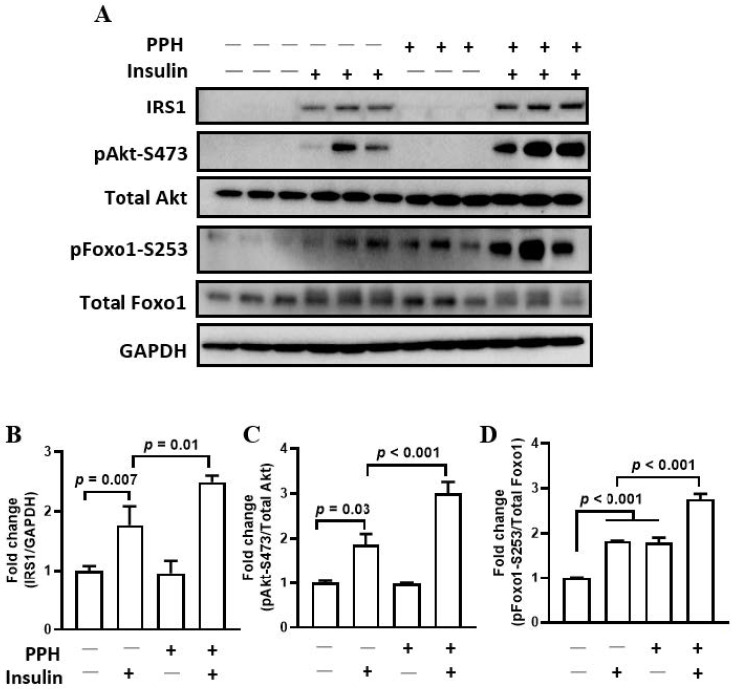
**PPH-activated insulin signaling in the AML-12 cells.** (**A**) AML-12 cells were pre-treated with PPH for 1h, followed by the co-treatment with 10 μM of insulin for 1 h. The total protein of the cells was extracted and immune-blotted with the antibodies against IRS1, pAkt-S473, total Akt, pFoxo1-S253, total Foxo1 or GAPDH. (**B**–**D**) The protein band of IRS1 was quantified and normalized to GAPDH. The protein bands of pAkt-S473 and pFoxo1-S253 were quantified and normalized to the corresponding total form. Data are expressed as mean ± SEM of at least 3 independent experiments.

**Figure 6 ijerph-19-10254-f006:**
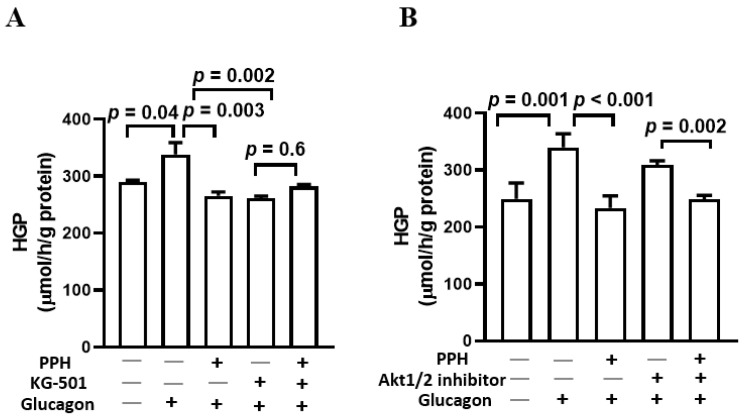
**PPH suppressed glucose production in the AML-12 cells via the glucagon signaling but not the insulin signaling**. AML-12 cells were cultured in glucose and phenol red-free DMEM containing sodium pyruvate and sodium lactate and pre-treated PPH and 10 μM of KG-501 (**A**) or 1 μM of Akt1/2 inhibitor (**B**) for 1 h, followed by the co-treatment with glucagon (100 nM) for 6 h. The concentration of glucose produced by the cells was normalized by the total protein content of the cells. Data are expressed as mean ± SEM of 5 independent experiments.

## Data Availability

Not applicable.

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
