# Peer review of "Pea Protein-Derived Peptides Inhibit Hepatic Glucose Production via the Gluconeogenic Signaling in the AML-12 Cells"

_ijerph, 2022, doi:10.3390/ijerph191610254_

Round 1

Reviewer 1 Report

I congratulate the authors of the article "Pea protein-derived peptides inhibit hepatic glucose production via the gluconeogenic signaling in the AML-12 cells" for their excellent work.

The manuscript is very well written and the English does not need corrections.

As a suggestion, the conclusion section could be improved. The way it's written doesn't value study.

Below are some other suggestions:

Lines 43 – 44: "compared with the synthetic hypoglycemic drugs, natural compounds have been considered safer alternatives due to the fewer side effects and relatively low production cost." Could the natural compounds referred to in this paragraph replace synthetic compounds? Or could they be used for disease prevention/control? Are there natural compounds that can be safely substituted for synthetic drugs? Can you cite some examples?

Line 138 - "P < 0.05 was considered to be significant.3." Replace P with p. I didn't understand ".3."

Figure 1A is not legible.

Author Response

Please find our response in the attachment. 

Reviewer 2 Report

Authors evaluate the in vitro activity of PPH on hepatic glucose production using the AML-12 cells. The study includes the peptide profiling of PPH, although, as a limitation, they did not evaluate the effect of single peptides or subfractions of this PPH, but of the complete hydrolyzate, which makes it difficult to attribute the effect to a specific molecule. However, experiments are well designed and results are interesting. Have you considered this analysis of peptides or subfractions?

On the other hand, this reviewer has a couple of comments:

 - Figure 4C and D: Authos menthioned " the glucagon induced transcriptions of G6PC and PCK could be suppressed 181 by the PPH treatment, which was evident by the reduced mRNA levels of these glucone- 182 ogenic genes (line 181)". However, to  this reviewer, the figure does not show this glucagon-induced enhancement and repression by PPH. Are the figure captions correctly indicated?

- Line 244: Pre-adipocytes

Author Response

(The authors gave the same response as above.)
